# Differentially Expressed miRNA of Prostate Cancer Compared with Benign Prostatic Hyperplasia Tissues: VAMP Associated Protein B Could Be Used for New Targets and Biomarkers of Prostate Cancer

**DOI:** 10.3390/biomedicines13122922

**Published:** 2025-11-28

**Authors:** Jae Heon Kim, Ahrim Moon, Miho Song, Kwang Woo Lee, Su Min Seo, Hui Ji Kim, Luis Alfonso Pefianco, Kevin Andrean, Seongho Ryu, Yun-Seob Song

**Affiliations:** 1Department of Urology, Soonchunhyang University School of Medicine, Seoul 04404, Republic of Korea; piacekjh@hanmail.net (J.H.K.); miho@schmc.ac.kr (M.S.); 2Department of Pathology, Soonchunhyang University School of Medicine, Bucheon 14584, Republic of Korea; armoon@schmc.ac.kr; 3Department of Urology, Soonchunhyang University School of Medicine, Bucheon 14584, Republic of Korea; urolkw@schmc.ac.kr; 4Soonchunhyang Institute of Medi-bio Science (SIMS), Soonchunhyang University, Cheonan 31151, Republic of Korea; niceolvia@naver.com (S.M.S.); khjee1932@gene2us.com (H.J.K.); 5Department of Integrated Biomedical Science, Soonchunhyang University, Cheonan 31151, Republic of Korea; luispefianco@sch.ac.kr (L.A.P.); kevinandrean@sch.ac.kr (K.A.)

**Keywords:** prostate cancer, benign prostate hyperplasia, Micro RNA, the protein vesicle-associated membrane protein-associated protein B

## Abstract

**Background/Objectives**: This NGS-based study sought to identify novel molecular markers for prostate cancer by comparing miRNA expression in cancer and benign prostatic hyperplasia (BPH) tissues. **Methods**: Using high-throughput sequencing and stringent statistical criteria, the study identified eleven significantly dysregulated miRNAs (five downregulated, six upregulated) that differentiate the two conditions. Enrichment analyses linked these miRNAs to several key cancer-associated pathways, including PI3K–Akt and ErbB signaling. **Results**: Crucially, the protein vesicle-associated membrane protein-associated protein B (VAPB) was pinpointed as a central hub, regulated by three downregulated miRNAs (miR-143-3p, miR-221-3p, and miR-222-3p). Since VAPB has not been widely studied in prostate cancer, it represents a promising, novel candidate for both diagnosis and therapeutic targeting. **Conclusions**: Our NGS-based analysis revealed a distinct miRNA expression signature that differentiates prostate cancer from BPH. The downregulation of several tumor-suppressive miRNAs (with concomitant upregulation of oncogenic miRNAs) in prostate cancer may contribute to malignancy—including the de-repression of novel targets like VAPB, which we identify as a promising new biomarker and therapeutic target.

## 1. Introduction

Prostate cancer is one of the most common malignancies in men and a major contributor to cancer-related morbidity and mortality worldwide [1]. Localized disease is often indolent with excellent survival [2,3], but prognosis worsens substantially once progression or metastasis occurs [4,5]. Differentiating aggressive tumors from indolent disease, and distinguishing malignancy from benign prostatic hyperplasia (BPH), remains a major clinical challenge. Because BPH frequently causes elevated prostate-specific antigen (PSA) levels and overlapping urinary symptoms, current diagnostic tools such as PSA lack specificity, emphasizing the need for more reliable molecular markers.

MicroRNAs (miRNAs) are small noncoding RNAs (~20–24 nucleotides) that regulate gene expression post-transcriptionally by binding target mRNAs to inhibit translation or promote degradation [6,7]. They are critical regulators of diverse biological processes including proliferation, differentiation, apoptosis, immune regulation, and metastasis [8]. A substantial proportion of human genes are under miRNA control, reflecting their broad biological influence.

Prostate cancer is highly heterogeneous at the molecular level [9]. Identifying disease- and stage-specific biomarkers is therefore attractive for improving diagnosis, prognostication, and treatment selection. Compared with protein or mRNA markers, miRNAs are stable in tissue and body fluids, and exhibit tumor-specific expression patterns, making them strong candidates for biomarker development. Indeed, prostate cancer–specific miRNA signatures have shown utility in distinguishing BPH from cancer and in predicting tumor aggressiveness [10,11]. Numerous miRNAs are aberrantly expressed in prostate cancer relative to normal prostate cells [12], contributing to tumorigenesis by modulating key gene networks [13,14]. Loss of tumor-suppressive miRNAs or upregulation of oncogenic miRNAs can drive malignant transformation and disease progression [10,14,15]. Importantly, miRNAs are detectable not only in tissues but also noninvasively in blood and urine [16], and advances in detection technologies further support their clinical utility [17].

Despite increasing interest, relatively few studies have compared miRNA expression directly between prostate cancer and BPH tissues. Most have examined cancer versus normal tissue or circulating miRNAs, limiting insights into malignancy-specific changes within the prostate. To address this gap, we performed next-generation sequencing (NGS) of prostate cancer and BPH tissues to identify differentially expressed miRNAs. Through bioinformatic analyses of predicted targets and pathways, we also explored their functional roles, with the aim of discovering novel biomarkers and potential therapeutic targets in prostate cancer.

## 2. Materials and Methods

### 2.1. Patient Recruitment and Sample Collection

This study analyzed prostate tissue samples from fourteen patients: nine with histologically confirmed prostate adenocarcinoma and five BPH. All cancer tissues were obtained from radical prostatectomy specimens, while BPH samples were collected from transurethral resection or simple prostatectomy procedures performed at Soonchunhyang University Hospitals in Seoul and Bucheon (January 2002–December 2012). Formalin-fixed paraffin-embedded (FFPE) blocks were prepared, and representative tumor or benign regions were identified by pathologists on hematoxylin–eosin (H&E)-stained slides. Corresponding FFPE blocks were used for RNA extraction. Clinical and pathological data—including age, serum PSA, Gleason grade, clinical risk category, and tumor stage—were retrieved from medical records. The study was approved by the Institutional Review Boards of Soonchunhyang University Seoul Hospital (IRB No. 2017-02-002) and Bucheon Hospital (IRB No. 2017-03-004), and written informed consent was obtained from each patient prior to sample collection.

Total RNA, including small RNA fractions, was isolated from 5 to 10 μm FFPE tissue sections using the miRNeasy Mini Kit (Qiagen, Hilden, Germany) according to the manufacturer’s instructions. RNA was eluted in 20 µL RNase-free water, and its yield and integrity were assessed with a NanoDrop spectrophotometer and agarose gel electrophoresis. A schematic of the collection and extraction workflow is shown in Figure 1.

### 2.2. Library Preparation and miRNA Sequencing

Small RNA libraries were prepared with the SMARTer smRNA-Seq Kit for Illumina (Takara Bio, Kusatsu, Japan), using 10 ng of total RNA per sample. After adapter ligation and reverse transcription, cDNA was amplified by PCR and fragments (~150 bp, including adapters) were purified by gel extraction. Library quality and size distribution were confirmed using the Agilent 2100 Bioanalyzer with a High Sensitivity DNA kit, and concentrations were quantified by qPCR (KAPA Biosystems, Wilmington, MA, USA). Libraries were pooled in equimolar amounts and sequenced on an Illumina HiSeq 2500 system in high-output mode to generate 101-base single-end reads. Image processing and base calling were performed with Illumina’s real-time analysis software, followed by demultiplexing and quality control. Raw data were deposited in the NCBI Gene Expression Omnibus (GSE179961). For miRNA annotation, high-quality reads were aligned to the human reference genome and compared against miRBase v21 using the miRDeep2 pipeline, enabling both known miRNA quantification and discovery. Reads with low base-call quality (Phred score < 20), shorter than 18 nucleotides, or containing ambiguous adapter sequences were removed. All 14 libraries were sequenced in a single run to minimize batch effects; no significant batch effect was observed in the data, so no batch correction was necessary. For miRNA identification, the high-quality reads were mapped to the human reference genome (GRCh38) and aligned with known miRNA sequences from miRBase v21 using the miRDeep2 pipeline (version 2.0.0.8).

### 2.3. Differential miRNA Expression Analysis

Differential expression between prostate cancer and BPH groups was assessed with the DESeq2 package in Bioconductor. In total, 11 differentially expressed miRNA were analyzed. The 5 down-regulated mRNA were miR-221-3p, miR-143-3p, miR-222-3p, miR-455-3p, miR-152-3p. The 6 up-regulated mRNA were miR-10524-5p, miR-1248, miR-9901, miR-663a, miR-142-3p, miR-4449 (Table 1). Low-abundance miRNAs (absent in >50% of samples) were excluded to reduce noise. Raw counts were normalized using the median-of-ratios method. Log2 fold changes and Wald tests (negative binomial model) were calculated for each miRNA. Significance was defined as fold change ≥ 2 with a Benjamini–Hochberg FDR < 0.05. Hierarchical clustering of significant miRNAs (Euclidean distance, complete linkage) was performed on normalized data, and results were visualized in a heatmap (Figure 2).

### 2.4. Target Prediction and Pathway Enrichment

Predicted mRNA targets of dysregulated miRNAs were identified using TargetScan 7.2 [18,19]. Conserved targets were then subjected to functional analysis. KEGG pathway enrichment was performed using DAVID and the KEGG PATHWAY database, with multiple-testing adjusted *p* < 0.05 considered significant. In parallel, enrichment of Disease Ontology (DO) terms were assessed to identify disease categories associated with the predicted gene sets. Pathways and disease terms enriched among targets of downregulated miRNAs are summarized in Figure 3.

### 2.5. Statistical Analysis

Clinical and demographic variables were compared between prostate cancer and BPH groups. Continuous variables were analyzed using Student’s *t*-test or the Mann–Whitney U test, and categorical variables using chi-square or Fisher’s exact test. Data are reported as mean ± standard error (SE). A two-sided *p* < 0.05 was considered statistically significant. Analyses were conducted with SPSS version 25.0 (IBM, Armonk, NY, USA) and R (R Foundation, Vienna, Austria).

#### TCGA-PRAD Public Dataset Analysis

To validate the differential miRNA expression findings, the TCGA-PRAD miRNA-seq dataset was utilized and accessed from the Genomic Data Commons (GDC) portal (https://portal.gdc.cancer.gov/, accessed on 24 November 2025) via the TCGAbiolinks R package (version 2.30.4) (Colaprico et al., 2016) [20]. To ensure comparability with the clinical characteristics of the patients, the dataset was stratified by Gleason scores, particularly the samples with a Gleason score of 6 and 7. This was represented by 275 primary tumor samples and 45 solid normal tissue samples from the dataset.

Differential expression of miRNAs was performed by comparing the primary tumor and normal tissue samples using the Bioconductor DESeq2 package (version 1.42.1). To enhance data quality and reliability a similar data filtering step was performed. In which, the miRNAs that were not detected (zero counts) in more than 50% of the samples were excluded from the analysis. Read counts were then normalized across the samples using the median-of-ratios method in DESeq2. This approach enabled validation of the identified miRNAs in a larger and independent cohort.

To further investigate biological functions, comprehensive functional enrichment analyses were performed. This was performed by leveraging established public dataset for further validation of the enriched pathway of the miRNA target genes. Gene ontology (GO) enrichment analysis was performed using the clusterProfiler package (version 4.10.1) with the GO.db annotation database (version 3.18.0). Kyoto Encyclopedia of Genes and Genomes (KEGG) pathway enrichment analysis was performed using clusterProfiler with the org.Hs.eg.db annotation package (version 3.18.0) to identify signaling pathways associated with the miRNAs. Disease Ontology (DO) enrichment analysis was performed using the DOSE package (version 3.28.2) to identify disease terms related with the target genes.

## 3. Results

### 3.1. Patient Characteristics

Fourteen prostate tissue samples (9 prostate cancer and 5 BPH) were analyzed. Clinical information is summarized in Table 2. Mean age did not differ significantly between groups (70.6 ± 1.2 vs. 68.6 ± 2.5 yr, *p* > 0.05). All prostate cancer cases were adenocarcinomas with a mean preoperative PSA of 22.8 ± 8.8 ng/mL. Most were early stage, with mean Gleason grade group 2.4 ± 0.6 and pathological T stage 1.6 ± 0.2, corresponding predominantly to pT2 tumors. The average NCCN risk score was 1.6 ± 0.2, indicating mainly low- to intermediate-risk disease.

### 3.2. Differential miRNA Expression

NGS-based profiling identified 11 miRNAs significantly dysregulated in prostate cancer compared with BPH (|fold change| ≥ 2, FDR < 0.05; Figure 2). Five miRNAs (miR-143-3p, miR-152-3p, miR-221-3p, miR-222-3p, miR-455-3p) were downregulated, while six (miR-10524-5p, miR-1248, miR-9901, miR-663a, miR-142-3p, miR-4449) were upregulated. A volcano plot illustrates the magnitude and significance of changes (Figure 2A), and hierarchical clustering of these 11 miRNAs clearly separated prostate cancer from BPH samples (Figure 2B). Figure 2B depicts a volcano plot of the miRNA expression changes, with the x-axis representing log_2_ fold changes and the y-axis showing statistical significance (−log_10_ FDR). The 11 miRNAs meeting our significance criteria (FDR < 0.05 and fold-change ≥ 2) are highlighted on the plot.

### 3.3. Pathway and Disease Ontology Analysis

Predicted targets of the five downregulated miRNAs were enriched for multiple cancer-related pathways, including PI3K–Akt, ErbB, EGFR inhibitor resistance, chemical carcinogenesis, and the “microRNAs in cancer” pathway (Figure 3A). These included MicroRNAs in cancer, ErbB signaling pathway, PI3K-Akt signaling pathway, EGFR tyrosine kinase inhibitor resistance, and chemical carcinogenesis—receptor activation. Several metabolic and disease pathways were also enriched, such as cholesterol metabolism and neurological disorder pathways like amyotrophic lateral sclerosis (ALS) and pathways of neurodegeneration. DO analysis confirmed enrichment of “prostate cancer” among predicted targets, particularly for miR-221-3p, miR-222-3p, and miR-455-3p (Figure 3B). Additional enriched terms included glioma, sarcoma, osteoporosis, and Huntington’s disease.

### 3.4. miRNA–mRNA Interaction Network

Network analysis of the five downregulated miRNAs identified several oncogenic targets, with VAPB (VAMP-associated protein B) emerging as a central hub regulated by miR-143-3p, miR-221-3p, and miR-222-3p (Appendix A and Figure 4). The interaction network for these three miRNAs is shown in Figure 4B. In Figure 4, nodes represent either a downregulated miRNA or one of its predicted target genes, and edges denote the regulatory interactions; shared target genes (including VAPB) regulated by multiple miRNAs appear as hub nodes in the network. Other notable targets included ERBB3, NRAS, and MDM4 for miR-152-3p (shown in Figure 4A), and CDKN1B, KIT, and SOCS3 for the miR-221/222 cluster (also shown in Figure 4B). The network for miR-455-3p is shown in Figure 4C. The convergence on VAPB suggests that its de-repression may contribute to prostate cancer biology, highlighting it as a potential novel biomarker and therapeutic target.

## 4. Discussion

In this study, we utilized next-generation sequencing to examine microRNA (miRNA) expression profiles in prostate cancer (PCa) and benign prostatic hyperplasia (BPH) tissues. We identified eleven miRNAs that were differentially expressed, with five downregulated and six upregulated in PCa compared to BPH (Table 1). These results provide insights into molecular differences between malignant and benign prostatic tissues. Notably, several of the dysregulated miRNAs have previously been implicated in cancer, supporting the biological relevance of our findings. To our knowledge, this represents one of the few studies specifically comparing miRNA expression between BPH and PCa, highlighting potential diagnostic biomarkers. Additionally, integrating target prediction and pathway analyses allowed us to propose mechanisms by which these miRNAs may contribute to PCa pathogenesis, including identifying VAPB as a novel oncogenic candidate.

Among the downregulated miRNAs in PCa (miR-143-3p, miR-152-3p, miR-221-3p, miR-222-3p, miR-455-3p), many have recognized tumor-suppressor functions. For example, miR-221/222 have been extensively studied, showing context-dependent roles. Goto et al. reported that miR-221/222 are decreased in castration-resistant prostate cancer, acting as tumor suppressors with lower expression linked to disease progression [21,22]. Similarly, Fuse et al. identified miR-222 and miR-31 as tumor-suppressive, with their loss promoting PCa via specific pathways [23]. Conversely, studies by Song et al. and Yang et al. showed miR-221/222 can act as oncogenes, promoting migration and invasion in PCa cell lines, while their inhibition reduced proliferation and induced apoptosis [21,24]. In our cohort of primary PCa tissues, miR-221-3p and miR-222-3p were downregulated relative to BPH, supporting a tumor-suppressive role in early-stage PCa. This duality emphasizes the context- and stage-specific effects of miR-221/222.

miR-143-3p, another downregulated miRNA, is a well-known tumor suppressor. Previous studies demonstrate that restoring miR-143 inhibits PCa cell growth and enhances docetaxel sensitivity by targeting oncogenes like KRAS [25]. Our results showing reduced miR-143-3p in PCa align with these findings, indicating its loss may promote tumorigenesis through activation of pro-proliferative pathways. miR-455-3p was also decreased in PCa samples. Zhao et al. (2017) reported miR-455-3p suppresses PCa cell growth by targeting eIF4E, a translation initiation factor that drives protein synthesis [26]. Loss of miR-455-3p could relieve this suppression, promoting tumor progression. miR-152-3p, another tumor-suppressive miRNA, regulates epigenetic modifiers such as DNMT1 and PTEN [27,28]. Downregulation of miR-152 may lead to DNA hypermethylation, PTEN loss, and PI3K-Akt pathway activation. Our findings of reduced miR-152-3p in PCa are consistent with these mechanisms, and pathway analysis confirmed enrichment of PI3K-Akt signaling.

In contrast, the upregulated miRNAs (miR-10524-5p, miR-1248, miR-9901, miR-663a, miR-142-3p, miR-4449) may act as oncomiRs. Some, such as miR-10524-5p and miR-9901, are novel and uncharacterized, while others have known oncogenic roles. miR-1248, upregulated in our PCa tissues, has been implicated in gliomas as an oncomiR [29]. miR-663a promotes castration-resistant phenotypes and correlates with aggressive clinical features in PCa [30]. miR-142-3p targets the tumor suppressor FOXO1, promoting proliferation [31], and miR-4449 may influence STAT3 signaling by targeting SOCS3, as suggested in other cancers [32]. The differential expression of these miRNAs suggests they could be novel biomarkers or contributors to PCa progression.

Integrative pathway and network analyses revealed that downregulated miRNAs converge on PI3K-Akt and ErbB signaling pathways, critical in prostate oncogenesis. For instance, miR-152 targets ERBB3, and its downregulation may enhance ErbB/PI3K signaling. Disease ontology enrichment indicated that miR-221/222 and miR-455 target genes associated with PCa, validating our approach [22,23,26]. Unexpected links to ALS pathways were explained by the identification of VAPB, a known ALS-related gene, as a key miRNA target [33,34].

VAPB, an ER-resident protein involved in the unfolded protein response, vesicle trafficking, lipid transfer, and calcium homeostasis [33,34,35,36,37], emerged as a novel candidate in PCa. While extensively studied in neurodegeneration, VAPB has limited prior association with cancer. Recent evidence indicates VAPB supports proliferation in medulloblastoma cells [38]. In our study, miR-143-3p, miR-221-3p, and miR-222-3p downregulation may lead to VAPB upregulation, potentially aiding tumor survival under ER stress or metabolic stress. To support the relevance of VAPB in prostate cancer, we performed an in silico validation using public gene expression datasets. Analysis of The Cancer Genome Atlas prostate adenocarcinoma cohort (TCGA-PRAD) revealed that VAPB mRNA expression is significantly higher in prostate cancer tissues compared to normal prostate tissues. Similarly, examination of an independent gene expression microarray dataset from GEO indicated that VAPB transcript levels are elevated in prostate tumor samples relative to benign prostate samples. These in silico findings provide additional evidence that VAPB is upregulated in prostate cancer, consistent with the loss of its miRNA-mediated repression in tumors [39,40,41,42,43]. Mutations in VAPB in familial neurodegenerative disease ALS patients were identified [44], but its relevance to prostate cancer was identified via bioinformatic analysis of miRNA targets. ALS pathway” came from a disease ontology enrichment result in our analysis. Although VAPB is known for its involvement in amyotrophic lateral sclerosis [37], our identification of VAPB in prostate cancer stems purely from bioinformatic predictions and pathway enrichment.

Reduced miR-221/222 expression in primary PCa tissues were reported [45,46]. While miR-221/222 can function as oncogenes in advanced or castration-resistant PCa (they are often elevated in hormone-refractory or metastatic cases [45]), their downregulation in early-stage disease suggests a context-dependent role. Galardi et al., even characterize the miR-221/222 cluster as tumor-suppressive in early PCa [46]. This is consistent with prior studies showing that miR-221/222 are downregulated in primary prostate tumors and may act as context-dependent tumor suppressors in early disease [24,44,45], even though they have oncogenic roles in later-stage or androgen-independent prostate cancer [44,45,46].

Another key downregulated miRNA in our data is miR-143-3p. This miRNA is well-established as a tumor suppressor in several cancers, including prostate cancer. Previous studies have shown that miR-143 is frequently downregulated in prostate tumors and that restoring miR-143 can suppress cancer cell growth [47,48,49,50]. For instance, Xu et al. demonstrated that miR-143 inhibits prostate cancer cell proliferation and migration, and it can also enhance the sensitivity of prostate cancer cells to the chemotherapy drug docetaxel by targeting key oncogenes like KRAS [24]. Schaefer et al. (2010) and Kojima et al. (2014), who observed significant under-expression of miR-143 in prostate tumors relative to normal prostate tissue were reported [49,50]. Our finding of reduced miR-143-3p in prostate cancer tissue relative to BPH is consistent with those reports and reinforces the idea that loss of miR-143 may contribute to prostate tumorigenesis by unleashing pro-proliferative pathways such as the KRAS signaling pathway. Loss of the miR-143/145 cluster has been shown to enhance prostate cancer cell invasion and migration (by de-repressing targets like GOLM1) [50], consistent with our finding that miR-143-3p is downregulated in tumor tissue. This provides biological context that miR-143 loss may contribute to PCa progression, supporting our suggestion that it functions as a tumor-suppressive miRNA in this setting [50].

miR-152 is frequently downregulated in prostate cancer, which is in line with our data [51]. miR-152 normally targets oncogenic pathways (e.g., it can target DNA methyltransferase 1 and TGFα) and that its loss has been linked to increased proliferation, invasion, and therapy resistance in prostate cancer [52]. This expanded context makes clear that our observation of miR-152-3p downregulation is well-supported by the literature and may have functional consequences (de-repression of miR-152 targets like DNMT1 or FOXR2, which promote tumor aggressiveness). Notably, miR-152-3p has been reported to be frequently downregulated in prostate cancer tissues, often via promoter hypermethylation, and to function as a tumor suppressor by restraining oncogenic signaling (e.g., the DNMT1-driven epigenetic circuit) [51,52]. Our finding of miR-152-3p downregulation in PCa aligns with these reports and suggests that loss of miR-152 could contribute to aberrant DNA methylation and oncogene activation in prostate malignancy.

This study is the first to suggest a potential role for VAPB in PCa. Future studies should confirm its expression in prostate tissues, assess functional significance, and investigate mechanisms by which VAPB may contribute to tumor growth, such as interactions with androgen receptor signaling or lipid and calcium signaling. If validated, VAPB could serve as a biomarker or therapeutic target, especially in approaches exploiting ER stress. Consistent with this, analysis of independent datasets indicates that VAPB transcript levels are higher in primary prostate tumors than in benign or normal prostate tissues (as evidenced by TCGA data), supporting the idea that VAPB is indeed upregulated in prostate cancer.

Limitations include a small sample size (9 PCa, 5 BPH) and use of FFPE samples. Functional studies are needed to establish causality between miRNA changes and tumor progression. This small cohort size is a major limitation of our study and limits the statistical power and generalizability of the findings. Larger cohorts would help validate the consistency of the 11 miRNA signature we identified. Additionally, our analysis was cross-sectional and exploratory in nature; functional studies were beyond its scope. Nevertheless, the identified miRNA panel holds potential as a diagnostic tool, potentially measurable in tissue, blood, or urine. Therapeutically, miRNA replacement or inhibition strategies could target tumor-suppressive or oncogenic miRNAs, respectively.

## 5. Conclusions

In summary, our study delineates a miRNA signature differentiating PCa from BPH, highlighting known tumor-suppressors (miR-221/222, miR-143) and novel candidates (miR-1248, miR-10524-5p). Pathway enrichment emphasizes PI3K-Akt signaling, and the miRNA–target network identifies VAPB as a potentially critical gene in PCa. These findings expand molecular understanding of PCa and provide a foundation for miRNA-based diagnostic and therapeutic strategies. Further validation in larger cohorts and functional assays will be essential for clinical translation.

## Figures and Tables

**Figure 1 biomedicines-13-02922-f001:**
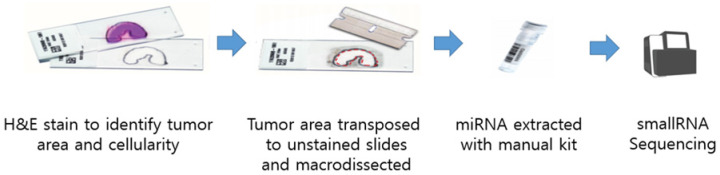
Schematic view of sample collection.

**Figure 2 biomedicines-13-02922-f002:**
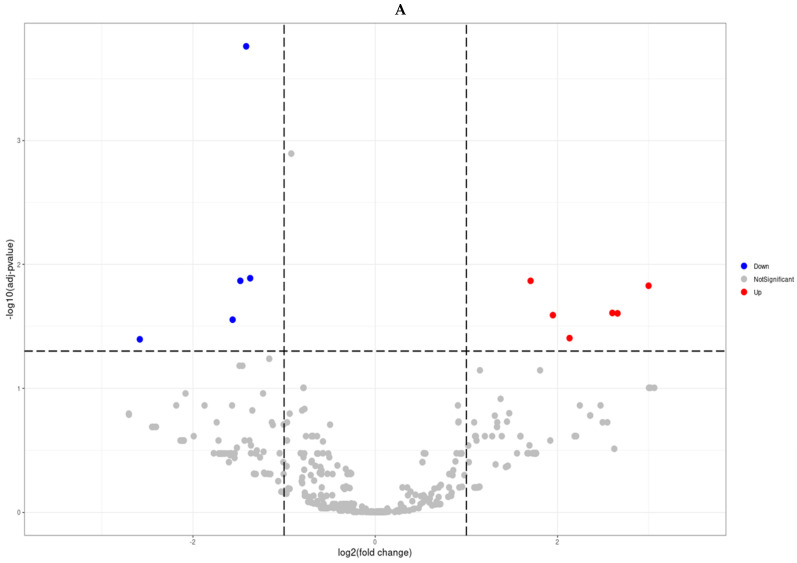
Differentially expressed miRNA analysis (Prostate cancer tissues vs. benign prostatic hyperplasia tissues). Volcano plots (**A**) and Hierarchical clustering (**B**) of microRNAs displaying differential differentially expressed miRNAs (Fold change ≥ 2 and FDR < 0.05).

**Figure 3 biomedicines-13-02922-f003:**
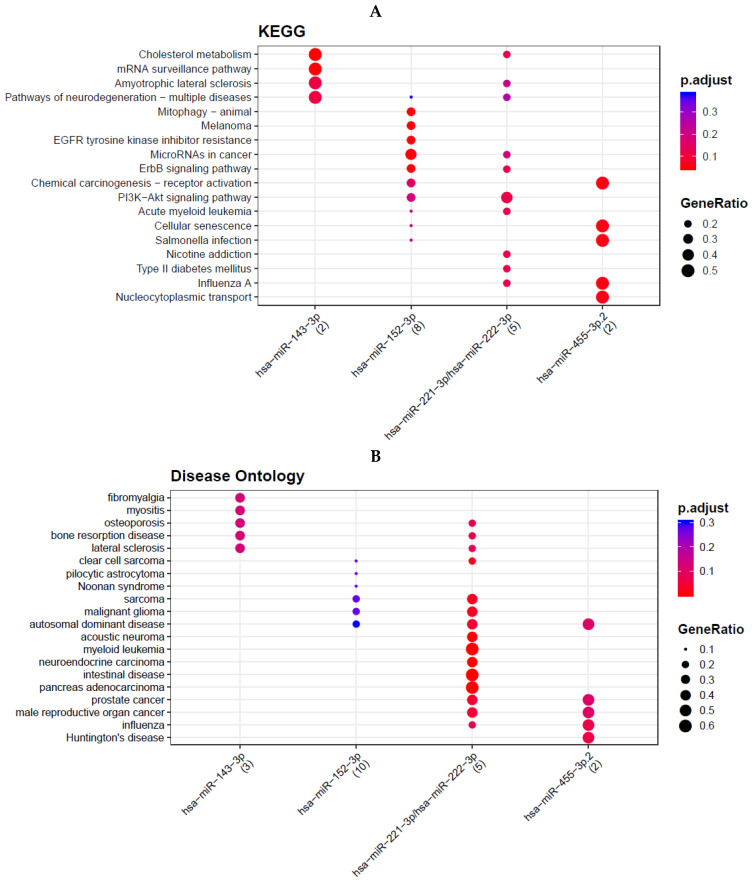
Five down-regulated mRNA. Pathway and disease ontology. Cholesterol me-tabolism, mRNA surveillance pathway, amyotrophic lateral sclerosis, pathways of neurodegen-eration-multiple disease, mitophagy, melanoma, EGFR tyro-sine kinase inhibitor resistance, mi-croRNAs in cancer, ErbB signaling pathway, chemical carcinogenesis-receptor activation, PI3K-Akt signaling pathway, acute myeloid leukemia, cellular senescence, salmonella infection, nicotine addiction, Type II diabetes mellitus, influenza A, nucleocytoplasmic transport. (**A**) Disease ontology analysis con-firmed enrichment of “prostate cancer” among predicted targets, particularly for miR-221-3p, miR-222-3p, and miR-455-3p (**B**).

**Figure 4 biomedicines-13-02922-f004:**
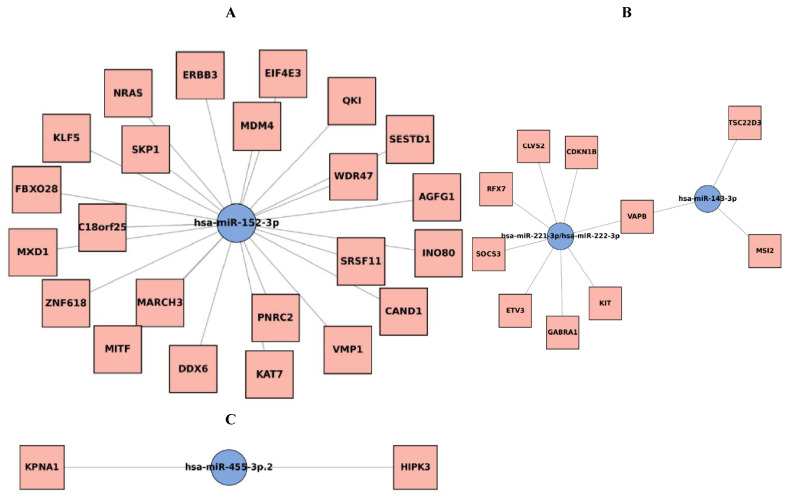
Five down-regulated microRNA genes interaction networks. (**A**): miR-152-3p. (**B**): miR-143-3p, miR-221-3p, miR-222-3p. VAPB genes interacted with miR-143-3p, miR-221-3p, miR-222-3p. (**C**): miR-455-3p. miRNA–target genes interaction networks for the downregulated miRNAs (Prostate cancer tissues vs. benign prostatic hyperplasia tissues). miRNA hubs are depicted as blue dots and targets as yellow dots. VAPB: VAMP (Vesicle-associated membrane protein)-associated protein B.

**Table 1 biomedicines-13-02922-t001:** Differentially expressed miRNA analysis.

miRNA(mature_precursor)	LogFC	*p* Value	FDR (Q Value)
has-miR-10524-5p_hsa-mir-10524	2.99852617	0.00018072	0.01487925
has-miR-1248_hsa-mir-1248	2.65823893	0.00040184	0.02487541
has-miR-9901_hsa-mir-9901	2.6009562	0.0003498	0.02468574
has-miR-663a_hsa-mir-663a	2.13141469	0.00087813	0.03943605
has-miR-142-3p_hsa-mir-142	1.94898106	0.00046858	0.02571969
has-miR-4449_hsa-mir-4449	1.70424794	0.00013744	0.01357934
has-miR-221-3p_hsa-mir-221	−1.37198402	7.86 × 10^−5^	0.01293899
has-miR-143-3p_hsa-mir-143	−1.4160143	3.51 × 10^−7^	0.00017321
has-miR-222-3p_hsa-mir-222	−1.48020016	0.00012267	0.01357934
has-miR-455-3p_hsa-mir-455	−1.56457919	0.00056648	0.02798432
has-miR-152-3p_hsa-mir-152	−2.58295908	0.00097907	0.04030488

**Table 2 biomedicines-13-02922-t002:** Clinical profiles of patients analyzed in differentially expressed miRNA.

	Benign Prostatic Hyperplasia	Prostate Cancer	*p* Value
Age	70.6 ± 1.2 (n = 5)	68.6 ± 2.5 (n = 9)	>0.05
PSA	1.7 ± 0.1 (n = 5)	22.8 ± 8.8 (n = 9)	<0.05
Gleason grade	NA	2.4 ± 0.6 (n = 9)	
Risk stratification	NA	1.6 ± 0.2 (n = 9)	
Pathologic T stage	NA	1.6 ± 0.2 (n = 9)	

## Data Availability

The data are not publicly available due to ethical restrictions. The data are; however, available from the corresponding author upon request.

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
