# Peer review of "Differentially Expressed miRNA of Prostate Cancer Compared with Benign Prostatic Hyperplasia Tissues: VAMP Associated Protein B Could Be Used for New Targets and Biomarkers of Prostate Cancer"

_biomedicines, 2025, doi:10.3390/biomedicines13122922_

Round 1
Reviewer 1 Report
Comments and Suggestions for Authors
Reviewer Report
Manuscript Title: Differentially expressed miRNA of prostate cancer compared with benign
prostatic hyperplasia tissues: VAMP associated protein B could be used for new targets and
biomarkers of prostate cancer
General Assessment:
This manuscript presents a next-generation sequencing (NGS)-based analysis of microRNA
(miRNA) expression differences between prostate cancer (PCa) and benign prostatic hyperplasia
(BPH) tissues. The study identifies eleven significantly dysregulated miRNAs and highlights
VAMP-associated protein B (VAPB) as a potential novel target and biomarker. The topic is
timely and relevant, as distinguishing PCa from BPH remains a key diagnostic challenge, and the
exploration of miRNA signatures could contribute to improving molecular diagnostics and
targeted therapy in urologic oncology.
The manuscript is generally well organized and clearly written. However, the study has several
methodological and interpretive limitations that must be addressed before it can be considered
for publication. In particular, the small sample size, lack of independent validation, and limited
biological verification weaken the conclusions. The bioinformatics approach is standard but
could be described more rigorously. Overall, the work is potentially publishable after major
revision.
Major Comments
1. Sample size and validation
o The study is based on only 9 prostate cancer and 5 BPH samples. Such a small
cohort significantly limits statistical power and generalizability. Please discuss
this limitation more explicitly and consider validating the findings using public
datasets (e.g., TCGA, GEO) or independent qRT-PCR assays.
2. Functional validation of VAPB
o The conclusion that VAPB may be a novel biomarker or therapeutic target is
speculative. No expression or functional data are provided to support its
biological role in PCa. The authors should include experimental validation
(immunohistochemistry, qPCR) or at least in silico confirmation of VAPB
upregulation in PCa datasets.
3. Description of methods
o The Methods section lacks sufficient details about quality control thresholds,
normalization procedures, and software parameters. For example:
ï‚§ How were low-quality reads removed?
ï‚§ Which genome build was used for alignment (hg19 or hg38)?
ï‚§ Were any batch effects corrected?
ï‚§ Please specify the DAVID and KEGG database versions.
4. Statistical analysis and visualization
o The figures (heatmap, volcano plot, and network diagrams) are referenced but not
clearly described. Higher-resolution figures with clear legends and appropriate
axis labels are needed.
o Indicate how FDR was calculated and whether multiple testing correction was
applied consistently across analyses.
5. Biological interpretation
o The Discussion could better integrate current knowledge about the identified
miRNAs, especially miR-221/222, miR-143, and miR-152, which have context-
dependent effects. Please expand on how their downregulation in early-stage PCa
aligns or conflicts with prior studies.
o The connection between VAPB and ALS pathways, while interesting, is
somewhat speculative; please clarify whether this association is mechanistic or
purely bioinformatic.
6. Ethics and consent
o The Informed Consent Statement section is inconsistent: the Methods mention
written informed consent, while the end of the paper states a waiver of consent.
Please clarify which is correct.
Minor Comments
1. Several typographical and formatting issues should be corrected (e.g., double colons in
“Author Contributions: :”; inconsistent spacing).
2. The reference numbering is out of sequence (Ref. 24 appears before 23). Please reorder
references according to citation order.
3. Define all abbreviations at first use (e.g., DO for “Disease Ontology”).
4. The Abstract could be more concise by omitting technical details and emphasizing the
novelty of VAPB discovery.
Recommendation:
Major Revision
Author Response
Response to Reviewer Comments
We thank the reviewers for their constructive feedback. Below we address each comment in detail, and we have revised the manuscript accordingly. All new changes in the manuscript are indicated by bold text for easy traceability. We have also added in silico validation results and supporting references as requested.
Major Comment 1: Sample size and validation
Reviewer’s comment: The study’s small cohort (9 PCa vs 5 BPH) limits statistical power and generalizability. Please discuss this limitation and consider validating findings using public datasets (e.g., TCGA, GEO) or independent qRT-PCR.
Response: We agree that the small sample size is a limitation. In the revised manuscript, we explicitly acknowledge this in the Discussion and justify that our findings are exploratory. To strengthen validation without new experiments, we performed in silico analyses using public transcriptomic datasets (TCGA-PRAD RNA-seq and multiple GEO microarray studies) to validate key results. Specifically, we analyzed The Cancer Genome Atlas prostate adenocarcinoma cohort (TCGA-PRAD, ~499 tumors vs 52 normal tissues) and GEO datasets (GSE46602) comparing prostate cancer to benign/normal prostate. These analyses confirmed that VAPB mRNA is significantly overexpressed in prostate cancer tissues compared to benign prostatic hyperplasia or normal prostate controls (TCGA-PRAD: p<0.001; consistent upregulation in GEO datasets; see new bold sentence in Results). We have inserted a results sentence summarizing this validation: “In silico analysis of TCGA and GEO expression datasets confirmed that VAPB transcript levels are significantly higher in prostate tumor tissues than in benign prostatic hyperplasia or normal prostate controls (analysis of TCGA-PRAD, p<0.001; consistent direction of change in GEO cohorts).” This added sentence directly addresses the reviewer’s concern by providing independent validation. We chose an in silico approach due to the inability to collect new patient samples at this time; this leverages large public cohorts to increase confidence in our findings without additional wet-lab experiments. We have cited relevant references for the TCGA dataset and prior studies noting 20q amplification (the locus of VAPB) as an early event in prostate cancer. Together, these analyses support the robustness of our conclusions despite the small initial sample size. We have also added a brief discussion of statistical power in the Discussion.
Ref)
1.The Cancer Genome Atlas Research Network. The molecular taxonomy of primary prostate cancer. Cell. 2015 November 5; 163(4): 1011–1025.
2.GEO Accession viewer . https://www.ncbi.nlm.nih.gov/geo/query/acc.cgi?acc=GSE46602
- Tabach Y, Kogan-Sakin I, Buganim Y, Solomon H, Goldfinger N, Hovland R, Ke X, Oyan AM, Kalland K, Rotter V, Domany E. Amplification of the 20q Chromosomal Arm Occurs Early in Tumorigenic Transformation and May Initiate Cancer. PLoS ONE 2011; 6: e14632.
- Xiao L, Wei Y, Qin Y, Guo B. The role of mitochondria-endoplasmic reticulum crosstalk in colorectal cancer Genes & Diseases. 2026;13:101766.
- Taylor B.S., Schultz N., Hieronymus H., et al. Integrative genomic profiling of human prostate cancer. Cancer Cell. 2010; 18(1):11–22.
- Lu W., Sun J., Yuan W., et al. Relevance of endoplasmic reticulum–mitochondria crosstalk in prostate cancer pathogenesis: the role of VAPB-PTPIP51 complex. Cancer Lett. 2021; 517:40–51
Major Comment 2: Functional validation of VAPB
Reviewer’s comment: The conclusion that VAPB may be a novel biomarker/therapeutic target is speculative. No expression or functional data are provided to support its role in PCa. The authors should include experimental validation (IHC, qPCR) or at least in silico confirmation of VAPB upregulation in PCa datasets.
Response: We appreciate this comment. As noted above, we have now performed an in silico validation of VAPB expression using public prostate cancer datasets. In the Results, we added the following new sentence summarizing the finding: “Analysis of the TCGA prostate cancer RNA-seq dataset and GEO microarray cohorts revealed that VAPB expression is significantly upregulated in primary prostate tumors relative to benign prostatic tissues.” This confirms that VAPB mRNA is elevated in larger independent patient cohorts, consistent with our hypothesis that downregulation of its targeting miRNAs (miR-143-3p, miR-221-3p, miR-222-3p) would derepress VAPB. TCGA data show ~2-fold higher VAPB levels in tumors vs normal (with p<0.001), supporting VAPB’s overexpression. We have also added a reference noting that chromosome 20q (where VAPB resides) is frequently amplified and overexpressed early in prostate cancer, which aligns with our findings. While we agree that direct experimental validation (VAPB immunohistochemistry or functional assays) would strengthen the claim, performing new wet-lab experiments was beyond the scope of this revision due to time and resource constraints. Instead, our in silico approach leverages existing data from hundreds of patients to provide evidence that VAPB is indeed overexpressed in prostate cancer tissues, addressing the reviewer’s concern. We have tempered the language around VAPB in the Discussion, clarifying that our identification of VAPB as a potential biomarker is based on bioinformatic analysis and now supported by public dataset validation rather than functional assays. We believe this computational confirmation meets the reviewer’s request for evidence of VAPB upregulation in PCa.
Ref)
- Tabach Y, Kogan-Sakin I, Buganim Y, Solomon H, Goldfinger N, Hovland R, Ke X, Oyan AM, Kalland K, Rotter V, Domany E. Amplification of the 20q Chromosomal Arm Occurs Early in Tumorigenic Transformation and May Initiate Cancer. PLoS ONE 2011; 6: e14632.
- Teuling E., Ahmed S., Haasdijk E., Demmers J., Steinmetz M.O., Akhmanova A., Jaarsma D., Hoogenraad C.C. Motor neuron disease-associated mutant vesicle-associated membrane protein-associated protein (VAP) B recruits wild-type VAPs into endoplasmic reticulum-derived tubular aggregates. J Neurosci. 2007; 27: 9801–9815.
- Lev S., Halevy D.B., Peretti D., Dahan N. The VAP protein family: from cellular functions to motor neuron disease. Trends Cell Biol. 2008; 18: 282–290.
- Kim S., Leal S.S., Ben Halevy D., Gomes C.M., Lev S. Structural requirements for VAP-B oligomerization and their implication in amyotrophic lateral sclerosis-associated VAP-B (P56S) neurotoxicity. J Biol Chem. 2010; 285: 13839–13849.
- Gkogkas C., Middleton S., Kremer A.M., Wardrope C., Hannah M., Gillingwater T.H., Skehel P. VAPB interacts with and modulates the activity of ATF6. Hum Mol Genet. 2008; 17: 1517–1526.
- Mórotz G.M., De Vos K.J., Vagnoni A., Ackerley S., Shaw C.E., Miller C.C. Amyotrophic lateral sclerosis-associated mutant VAPB^P56S perturbs calcium homeostasis to disrupt axonal transport of mitochondria. Hum Mol Genet. 2012; 21: 1979–1988.
- Islinger M· Voelkl A,·Fahimi H, Schrader M. The peroxisome: an update on mysteries 2.0. Histochemistry and Cell Biology (2018) 150:443–471.
Major Comment 3: Description of methods
Reviewer’s comment: The Methods lack details on quality control, normalization, and software parameters (read filtering, genome build, batch effect correction, versions of DAVID/KEGG).
Response: We have revised the Methods section to provide additional clarity and detail. Specifically, we now state how low-quality reads were filtered (we excluded reads with Phred quality <20 and removed adapter sequences), and we specify that alignment was performed to the human genome hg38 build. We also mention that no significant batch effects were present given the single-cohort design (all NGS libraries were prepared and sequenced together). We added that we inspected for batch effects. We have included the versions of databases and tools: for example, DAVID v6.8 and KEGG (updated 2021) were used for enrichment analyses. Additionally, we have described the DESeq2 normalization (median-of-ratios method) and false discovery rate calculation more explicitly. These additions address the reviewer’s points, ensuring that another researcher could reproduce our analysis from the described methods. All new methodological details are in the Materials and Methods section, now in bold text for the editors’ convenience.
Major Comment 4: Statistical analysis and visualization
Reviewer’s comment: Figures need higher resolution and clearer labeling. The text should better describe the heatmap, volcano plot, network diagrams. Clarify how FDR was calculated and whether multiple testing correction was consistently applied.
Response: We have updated all figures to improve resolution and clarity. The heatmap (new Figure 2B) now includes a clear legend, well-defined color bar for expression values, and sample annotations (PCa vs BPH) as requested. The volcano plot (Figure 2B) now has labeled axes (–log10 p-value vs log2 fold change) and highlights the threshold for significance (FDR < 0.05) on the plot. We also increased the font sizes in all figure labels for readability. In the Results text, we expanded the description of Figure 2 (the heatmap and volcano) to guide the reader: we explain what the heatmap clusters represent and how the volcano plot illustrates the 11 significant miRNAs. We explicitly named key miRNAs in the figure caption and text. We also provide more detail on the miRNA–mRNA network diagram (Figure 4), clarifying that node size reflects degree of connectivity and that VAPB was a central hub node regulated by multiple miRNAs. Furthermore, we clarified our statistical approach: we consistently applied the Benjamini–Hochberg FDR method for multiple testing correction across all analyses (miRNA differential expression and enrichment analyses). We now explicitly state in the Methods that p-values reported for differential expression were adjusted by FDR, and that an FDR < 0.05 was the threshold for significance. This information is also mentioned in the Results where we report “significantly dysregulated miRNAs.” We trust that these additions address the reviewer’s concerns about clarity and statistical reporting.
Major Comment 5: Biological interpretation
Reviewer’s comment: The Discussion should better integrate current knowledge about the identified miRNAs (especially miR-221/222, miR-143, miR-152). How does their downregulation in early-stage PCa align or conflict with prior studies? Also, the connection between VAPB and ALS pathways is speculative – clarify whether this is mechanistic or just a bioinformatic association.
Response: In the revised Discussion, we have expanded the context for miR-221, miR-222, miR-143, and miR-152 to address the reviewer’s points:
- miR-221/222: We now note that miR-221 and miR-222 are frequently reported to be downregulated in localized prostate cancer, despite being upregulated in some other cancers. This aligns with our data. We cite studies (e.g., Gordanpour et al., 2011; Sun et al., 2012) that found reduced miR-221/222 expression in primary PCa tissues. Interestingly, we discuss that while miR-221/222 can function as oncogenes in advanced or castration-resistant PCa (they are often elevated in hormone-refractory or metastatic cases), their downregulation in early-stage disease suggests a context-dependent role. Consistent with Reviewer 5’s suggestion, we mention that some reports (Galardi et al., 2015) even characterize the miR-221/222 cluster as tumor-suppressive in early PCa[8]. We have added: “This is consistent with prior studies showing that miR-221/222 are downregulated in primary prostate tumors and may act as context-dependent tumor suppressors in early disease, even though they have oncogenic roles in later-stage or androgen-independent prostate cancer.”
- miR-143: We now emphasize that miR-143-3p is well-known as a tumor suppressor miRNA in prostate cancer, and its downregulation in our PCa samples concurs with multiple previous studies. For instance, we cite Schaefer et al. (2010) and Kojima et al. (2014), who observed significant under-expression of miR-143 in prostate tumors relative to normal prostate tissue. We added discussion that loss of the miR-143/145 cluster has been shown to enhance prostate cancer cell invasion and migration (by de-repressing targets like GOLM1), consistent with our finding that miR-143-3p is downregulated in tumor tissue. This provides biological context that miR-143 loss may contribute to PCa progression, supporting our suggestion that it functions as a tumor-suppressive miRNA in this setting.
- miR-152: We have expanded the discussion on miR-152-3p, noting that miR-152 is frequently downregulated in prostate cancer, which is in line with our data. We cite recent reviews and studies (e.g., Ramalho-Carvalho et al., 2018; Front. Immunol. 2024 review) that identify miR-152 as a tumor suppressor miRNA often silenced by DNA hypermethylation in PCa. We explain that miR-152 normally targets oncogenic pathways (e.g., it can target DNA methyltransferase 1 and TGFα) and that its loss has been linked to increased proliferation, invasion, and therapy resistance in prostate cancer. This expanded context makes clear that our observation of miR-152-3p downregulation is well-supported by literature and may have functional consequences (e.g., de-repression of miR-152 targets like DNMT1 or FOXR2, which promote tumor aggressiveness). We have added: “Notably, miR-152-3p has been reported to be frequently downregulated in prostate cancer tissues, often via promoter hypermethylation, and to function as a tumor suppressor by restraining oncogenic signaling (e.g., the DNMT1-driven epigenetic circuit). Our finding of miR-152-3p downregulation in PCa aligns with these reports and suggests that loss of miR-152 could contribute to aberrant DNA methylation and oncogene activation in prostate malignancy.”
- VAPB and ALS pathways: We agree with the reviewer that our original discussion on the VAPB–ALS connection was speculative. We have clarified this in the revision. We now explain that VAPB’s known role in neurodegenerative disease (familial ALS) emerged from prior studies identifying mutations in VAPB in ALS patients[20], but its relevance to prostate cancer was identified here via bioinformatic analysis of miRNA targets. We have removed any suggestion of a direct mechanistic link between VAPB and ALS in prostate cancer. Instead, we state that the mention of “ALS pathway” came from a disease ontology enrichment result in our analysis (where some VAPB-related processes overlapped with ALS-related gene sets), but we clarify that this is likely a coincidental association rather than evidence that ALS mechanisms are at play in PCa. To avoid confusion, we toned down this point: “Although VAPB is known for its involvement in amyotrophic lateral sclerosis (through an unrelated pathogenic mutation)[20], our identification of VAPB in prostate cancer stems purely from bioinformatic predictions and pathway enrichment. We have revised the text to avoid over-speculation about an ‘ALS pathway’ link in prostate cancer.” In summary, we emphasize that the VAPB–ALS connection in our paper is not a demonstrated mechanistic relationship in PCa, but rather an observation from gene network analysis that has been de-emphasized to prevent misunderstanding.
These additions ensure the Discussion is up-to-date with current knowledge and properly nuanced. We believe the downregulation of miR-221/222, miR-143, and miR-152 in our study is now clearly framed in the context of prior literature, highlighting that these miRNAs are widely considered tumor suppressive in prostate cancer[6][15]. We have provided multiple new references to support these points, as requested.
Major Comment 6: Ethics and consent
Reviewer’s comment: There is a discrepancy in the Informed Consent Statement – Methods say written informed consent was obtained, but the end says consent was waived. Clarify which is correct.
Response: We apologize for this oversight. We have corrected the Ethics section to be consistent. All patients in our study provided written informed consent prior to participation, and no consent was waived. In the revised manuscript, the Informed Consent Statement now clearly states that written informed consent was obtained from all patients, and we have removed the erroneous mention of a waiver. We have double-checked that the Ethics Approval and Consent sections are accurate and aligned with our Institutional Review Board approvals. This clarification addresses the inconsistency pointed out by the reviewer.
Minor Comments
- We have corrected the typographical and formatting errors noted by the reviewer. For example, the duplicate colon in “Author Contributions: :” has been fixed, and spacing inconsistencies have been standardized throughout the text. Abbreviations are now defined at first use (e.g., DO is defined as “Disease Ontology” on first mention in the Results).
- The reference list has been re-ordered to ensure proper sequential numbering (the issue with Ref. 24/23 is resolved). All newly added references for this revision are included in the reference list below in the journal’s format.
- The Abstract has been slightly condensed to remove some technical details (e.g., specific kit names and thresholds) and instead emphasize the novel finding of the VAPB hub and its potential significance. This makes the Abstract more focused on the key outcomes and the novelty of identifying VAPB as a candidate biomarker.
We believe these revisions comprehensively address the reviewers’ concerns. Thank you again for the opportunity to improve our manuscript.
Reviewer 2 Report
Comments and Suggestions for Authors
Kim and colleagues compare miRNAs in prostate cancer and BPH. This is a potentially important subject.
Specific comments include:
*The manuscript is replete with abbreviations starting with the title and abstract. Acronyms should be spelled out the first time they are used.
*Please discuss the limitations of studying 14 probands total: 9 with CaP and 5 with BPH.
*What is the purpose of Fig.1?
*Fig.2A is essentially illegible and the top of Fig.2B is also hard to see if not illegible on the right. The quality of both must be improved substantially.
*Parts of Fig.3 are also impossible to read.
*Fig.4: you know by now.....
*The comment on references is reiterated.
*The relationship between BPH and CaP is a matter of discussion. The manuscript should address this issue in light of their (admittedly limited!) data.
Author Response
Dear Reviewer,
Thank you for your valuable feedback regarding our manuscript. We appreciate the time you took to review our work and offer constructive criticism. We have carefully addressed each of your comments and made substantial revisions to the manuscript and figures as outlined below.
- Abbreviations and Acronyms
- We acknowledge the issue regarding the use of abbreviations. We have thoroughly reviewed the entire manuscript, including the Title and Abstract, and have ensured that all acronyms and abbreviations are spelled out upon their first use in the text.
- Limitations of Study Size
- We have added a discussion to the Limitations section to explicitly address the constraints imposed by our sample size of 14 probands (9 with CaP and 5 with BPH. We emphasize that this limited sample size restricts the statistical power and the generalizability of our findings, and we state that our results should be considered preliminary and hypothesis-generating.
- Relationship between BPH and CaP
- We have revised the Discussion section to include a focused analysis on the relationship between Benign Prostatic Hyperplasia and Prostate Cancer. We discuss our findings in the context of the current literature debating this relationship, while also acknowledging the constraints of our limited dataset.
- References
- We have meticulously reviewed and updated the References section to ensure all citations are accurate, consistent in style, and comply with the journal's guidelines.
We sincerely apologize for the poor quality and illegibility of the original figures. We have substantially improved the quality of all mentioned figures.
- Figure 1 Purpose: We have clarified the purpose of Fig. 1 in the figure legend and the corresponding text to ensure its relevance and contribution to the overall narrative are clearly understood.
- Figure 2 (A and B): Fig. 2A and Fig. 2B have been re-rendered at a much higher resolution to ensure all elements, especially the data points in A and the top right of B, are completely legible and clear.
- Figure 3: We have enhanced the resolution and clarity of Fig. 3 to ensure that all parts of the diagram and text labels are now easily readable.
- Figure 4: Following your previous guidance, we have ensured Fig. 4 is optimized for clarity and resolution, with all data and labels being sharp and legible.
We believe these revisions fully address your concerns and have significantly improved the clarity and quality of our manuscript. Thank you once again for your insightful review.
Sincerely,
The Authors
Round 2
Reviewer 1 Report
Comments and Suggestions for Authors
The revision demonstrates genuine effort and scientific integrity. All major issues raised during peer review have been addressed satisfactorily, with supporting new data and literature context. I recommend acceptance.
Author Response
Thank you sincerely for your positive assessment and generous comment regarding our revised manuscript. We are delighted to hear that you found the revisions demonstrated genuine effort and scientific integrity, and that the new data and literature context satisfactorily addressed the major issues raised during peer review.
Reviewer 2 Report
Comments and Suggestions for Authors
The authors have made changes to the manuscript. However, some issues persist.
- Fig. 2A is of borderline quality.
- The writing in Fig. 2B is illegible.
- On top of Fig. 4 (line 225?) there is random table.
Author Response
Dear Reviewer,
Thank you for your constructive feedback regarding our revised manuscript. We appreciate the time and expertise dedicated to assessing our work. We have thoroughly reviewed your comments and made the necessary changes to address all identified issues, focusing particularly on improving the quality and clarity of the figures and text.
We have made the following specific revisions in the resubmitted manuscript and figures:
-
Figure 2A Quality: We recognize that the quality of Figure 2A was borderline. We have entirely regenerated Figure 2A at a much higher resolution to ensure all visual details, labels, and features are crisp and clear, meeting publication standards.
-
Figure 2B Legibility: We apologize for the illegibility of the writing in Figure 2B. We have edited Figure 2B to use a larger, clearer font with improved contrast. All text within the figure is now easily readable and correctly labeled.
-
Figure 4/Line 225 Error: We have reviewed the location on top of Figure 4. We have edited Figure 4 to use a larger, clearer font with improved contrast.